# WHERE REDUNDANCY LIVES: STAGE-AWARE BLOCK SALIENCY IN SKIP-CONNECTED MODELS

## ABSTRACT

Residual (skip-connected) architectures such as ResNets are widely used, yet the extent and structure of their inference-time redundancy remain unclear. We repurpose post-training block ablation as a diagnostic probe: we ablate residual blocks by replacing them with identity mappings, then measure the resulting accuracy drop on a small training "probe" slice, yielding a block-level saliency map that we evaluate out of sample on ImageNet. Across ResNet-50, our stage-aware analyses show that simple magnitude or energy proxies are weak or inconsistent predictors of, indicating that large activation does not imply importance; redundancy is better explained by low novelty relative to the skip path. We characterize structure using stage-wise distributions, and we assess practical trade-offs by one-shot identity replacement of those blocks with optional short finetuning, reporting realistic latency-accuracy behavior on CPU and GPU while preserving topology. The methodology is architecture-agnostic and readily extends to other modern skip-connected families (for example, ConvNeXt and ViT). These findings provide a simple, evidence-based way to localize redundancy, and to guide architecture-preserving simplifications at inference.

## 1 INTRODUCTION

Modern deep networks are substantially *overparameterized*: redundant capacity helps optimization and generalization, yet much of it may be unused or unnecessary at inference (Frankle & Carbin, 2019; Neyshabur et al., 2019). Residual (skip-connected) architectures such as ResNets (He et al., 2016) epitomize this trade-off. Identity shortcuts stabilize training and enable depth, but they also raise a basic post-hoc question: *after convergence, which residual blocks are still necessary for inference?*

We address this question with a simple, architecture-preserving **diagnostic**. For a trained residual block $B_i(x) = F_i(x) + x$, we construct an identity-ablated variant $B_i^{\text{abl}}(x) = x$ and measure the out-of-sample accuracy drop on ImageNet. The resulting one-number readout per block,

$$\delta_i = \alpha_{\text{base}} - \alpha_i^{\text{abl}},$$

constitutes a *necessity map* of the trained network (low $\delta_i \Rightarrow$ dispensable at inference). We compute selections on a small, fixed *probe* slice of the training data and evaluate on validation to avoid overfitting the diagnostic. The procedure alters neither topology nor training; it is a measurement tool, not an optimization method.

A widespread heuristic equates large activations with importance. Many pruning and scoring techniques lean on magnitude—norms, energy/gain, or first/second-order surrogates (Li et al., 2017; He et al., 2017; Molchanov et al., 2017; 2019; Lin et al., 2020; Ding et al., 2021; Shen et al., 2022). Our analyses show that in skip-connected networks this intuition is unreliable: **magnitude is not necessity**. On ResNet-50/ImageNet, simple magnitude or "energy" proxies (e.g., $\mathbb{E}\|y\|$, $\mathbb{E}\|y - x\|$, gain) are *weak or inconsistent* predictors of $\delta$ once we respect stage/depth. Global correlations can appear nontrivial, but stage-aware and stage-normalized statistics reveal that raw activation size overestimates importance under identity skips. A more faithful explanation of necessity is *novelty relative to the skip path*, not absolute magnitude.

We further examine how necessity evolves during training. Using checkpoints saved from a conventional ImageNet training, we compute $\delta$-maps at multiple epochs on the *same* probe indices and

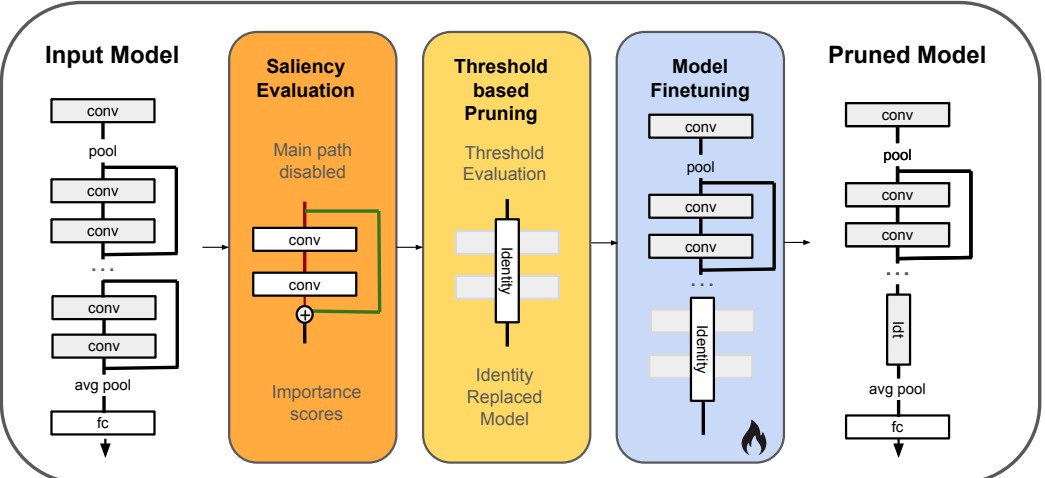

Figure 1: Overview of our post-training residual block pruning pipeline. The method consists of four stages: (1) saliency evaluation by ablating each residual block's main path to compute importance scores, (2) threshold-based pruning where low-saliency blocks are selected for removal, (3) identity-based replacement of the selected blocks, and (4) full-model finetuning to recover any accuracy lost from structural modifications. This pipeline operates entirely post-hoc and requires no modification to the original training process.

track their structure over time. Two robust patterns emerge: (i) later stages become increasingly prunable (lower median $\delta$), and (ii) the $\delta$ ranking stabilizes as optimization proceeds (higher Jaccard/Kendall agreement). Together with prior views of ResNets as ensembles of short paths (Veit et al., 2016) and results on stochastic layer skipping during training (Huang et al., 2016; Fan et al., 2019), these measurements yield a post-hoc account of redundancy: *it emerges with convergence*.

Finally, we translate the diagnostic into practice. Replacing the $K$ lowest-$\delta$ blocks with identity—optionally followed by a short recovery finetune—produces transparent latency–accuracy trade-offs on CPU and GPU while preserving the original computation graph. Because identity replacements keep interfaces intact, this approach is easy to deploy and extends naturally to other skip-connected families (e.g., ConvNeXt (Liu et al., 2022) and ViT (Dosovitskiy et al., 2021)).

**Contributions.**

- **Post-training diagnostic of block necessity.** Identity ablations yield $\delta$-maps that localize inference-time utility without re-training or topology changes.

- **Magnitude is not necessity.** Stage-aware analyses show activation energy/gain are weak or inconsistent predictors of $\delta$; necessity aligns better with novelty relative to the skip path.

- **Redundancy emerges during training.** Across epochs, later stages become safer to ablate and the $\delta$ ranking stabilizes, indicating redundancy formation with convergence.

- **Topology-preserving trade-offs.** One-shot identity replacement of low-$\delta$ blocks (optional short finetuning) yields realistic latency–accuracy curves on commodity hardware.

## 2 RELATED WORK

We situate our study within three strands: (i) *overparameterization* and why inference-time necessity can differ from training-time capacity; (ii) *skip connections and dynamic skipping*, which suggest resilience and potential redundancy in residual architectures; and (iii) *structured pruning and proxy scoring*, where magnitude- or curvature-based heuristics are commonly used. Our contribution is a

post-training, topology-preserving *diagnostic*: identity ablations that directly measure block necessity without re-optimization. This lens complements compression-oriented methods by prioritizing measurement over modification.

**Overparameterization and post-hoc necessity.** Modern deep networks are intentionally overparameterized: excess capacity improves optimization dynamics and often generalization, while leaving room for sparsity at inference (Frankle & Carbin, 2019; Neyshabur et al., 2019). A large body of work studies when and why small subnetworks can match the performance of their dense parents (e.g., lottery tickets and re-initialization effects) (Frankle & Carbin, 2019). These perspectives speak to *capacity* during training, but they do not directly measure which components of a *trained* model are still necessary for inference. Our work is complementary: we provide a post-training, topology-preserving *diagnostic* that quantifies block necessity via identity ablations, without re-optimization.

**Skip connections, redundancy, and dynamic skipping.** Residual networks introduce identity shortcuts that enable extremely deep models by stabilizing gradient flow (He et al., 2016). Several lines of evidence suggest structured redundancy in these architectures. ResNets can behave like ensembles of short paths, implying that not all blocks are equally critical (Veit et al., 2016). Training-time stochastic skipping (Stochastic Depth; LayerDrop) shows that randomly omitting blocks during optimization often preserves accuracy (Huang et al., 2016; Fan et al., 2019). Learned, input-dependent skipping (conditional computation) further gates blocks at inference for efficiency, e.g., SkipNet and BlockDrop (Wang et al., 2018; Wu et al., 2018). These approaches demonstrate resilience to missing computation *during training* or under learned policies. In contrast, our goal is a *post-hoc, measurement-driven* account: we assess which blocks remain necessary after convergence by replacing transformations with identity and reading out the accuracy drop.

**Structured pruning and magnitude/proxy scoring.** A parallel literature removes channels or blocks to shrink models. Many criteria are based on magnitude: $\ell_1/\ell_2$ norms of filters or activations (Li et al., 2017; He et al., 2017; Liu et al., 2017), rank/entropy of features (Lin et al., 2020), or first/second-order surrogates such as Taylor scores and Fisher-based importance (Molchanov et al., 2017; 2019). Later works specialize these ideas for residual networks (e.g., ResRep, LayerPrune) and typically couple pruning with finetuning to recover accuracy (Ding et al., 2021; Shen et al., 2022). While effective for *compression*, these methods often conflate activation *magnitude* with functional *necessity*—a mismatch that can be amplified by identity skips. Our experiments make this precise: when depth and stage are respected, simple magnitude or "energy" proxies (e.g., $\mathbb{E}\|y\|$, $\mathbb{E}\|y - x\|$, gain) are weak or inconsistent predictors of the post-training necessity measured by identity ablations. In other words, *magnitude is not necessity* in skip-connected models.

## 3 METHODOLOGY

### 3.1 SETTING AND DATA SPLITS

We study trained residual networks with blocks $B_i(x) = F_i(x) + x$, where $F_i$ is the transformation branch and $x$ the skip input. Blocks are grouped into stages $\mathcal{S}_1, \ldots, \mathcal{S}_m$ (e.g., the standard four ResNet stages). We use two disjoint sets: **PROBE** (a fixed subset of the training set) for computing selections/statistics, and **EVAL** (the validation set) for reporting accuracy. Unless stated otherwise, selections are computed once on probe and evaluated out of sample on eval.

### 3.2 POST-TRAINING BLOCK NECESSITY VIA IDENTITY ABLATION

For each block $i$, we form an identity-ablated variant that disables only the main branch:

$$B_i^{\mathrm{abl}}(x) = x \qquad \text{(skip preserved; any downsample projection kept).} \qquad (1)$$

Let $\alpha_{\mathrm{base}}$ be Top-1 accuracy of the unmodified model on the probe and $\alpha_i^{\mathrm{abl}}$ the accuracy of the model where only block $i$ is identity-ablated. The *necessity* of block $i$ is the accuracy drop

$$\delta_i = \alpha_{\mathrm{base}} - \alpha_i^{\mathrm{abl}}. \qquad (2)$$

Low $\delta_i$ indicates dispensability at inference.

**Identity replacement and finetuning.** Given a budget $K$, we replace the $K$ lowest-$\widehat{\delta}$ blocks *globally*. After replacement, we *finetune* the model to recover accuracy; the training recipe (number of epochs, learning-rate peak/decay) is *experiment-dependent* and reported alongside each result.

### 3.3 MAGNITUDE PROXIES AND THE "MAGNITUDE IS NOT NECESSITY" CLAIM

To test whether simple activation magnitude predicts necessity, we compute proxy scores with forward hooks on the probe:

$$
\begin{aligned}
\text{Output energy:} \quad & \mathbb{E}_{x \sim \text{PROBE}}[\|y\|_F], \\
\text{Residual energy:} \quad & \mathbb{E}_{x \sim \text{PROBE}}[\|y - x\|_F], \\
\text{Input energy:} \quad & \mathbb{E}_{x \sim \text{PROBE}}[\|x\|_F], \\
\text{Gain:} \quad & \mathbb{E}_{x \sim \text{PROBE}}\left[\frac{\|y\|_F}{\|x\|_F + \varepsilon}\right], \qquad \varepsilon > 0.
\end{aligned}
\tag{3}
$$

Where $B_i$ denotes residual block $i$. For an input activation tensor $x$ entering $B_i$, the block output is $y = B_i(x) = x + F_i(x)$, where $F_i$ is the transformation. We measure "energie" with forward hooks: (i) *Output energy* uses the post-block tensor $y$; (ii) *Residual energy* uses the residual branch $F_i(x) = y - x$; (iii) *Input energy* uses the pre-block tensor $x$. The norm $\|\cdot\|_F$ is the Frobenius (elementwise $\ell_2$) norm over all channels and spatial/sequence positions of a sample; expectations $\mathbb{E}[\cdot]$ are taken as simple averages over the fixed probe slice. The *gain* is a scale-normalized proxy $\mathbb{E}[\|y\|_F/(\|x\|_F + \varepsilon)]$ with a small $\varepsilon > 0$ for numerical stability.

We then relate these proxies to $\delta$ using the following statistics:

1. **Global Spearman** ($\rho$): rank correlation between a proxy and $\delta$ over *all* prunable blocks.

2. **Per-stage Spearman** ($\rho$): rank correlation computed *within* each stage $\mathcal{S}_s$ separately. Tests whether the proxy tracks necessity when depth/scale are held fixed.

3. **Stage-normalized ranks**: rank-transform each variable *within* its stage, pool the ranks across stages, and recompute Spearman. Provides a single summary that removes between–stage scale effects.

### 3.4 EMERGENCE DURING TRAINING: RANKING STABILITY

Our central question is whether the *ordering* of block necessity becomes predictable as training proceeds. For each checkpoint $e \in \mathcal{E}$ we form the necessity vector $\mathbf{d}^{(e)} = (\delta_1^{(e)}, \ldots, \delta_N^{(e)})$ over the $N$ prunable blocks (Sec. 3.2). We then assess stability in two complementary ways:

**Set stability of the "least-$K$".** Let $\text{leastK}^{(e)} \subseteq \{1, \ldots, N\}$ denote the indices of the $K$ lowest entries of $\mathbf{d}^{(e)}$. We quantify overlap with a checkpoint epoch $e^\star$ using the Jaccard index

$$
J_K(e, e^\star) = \frac{\left|\text{leastK}^{(e)} \cap \text{leastK}^{(e^\star)}\right|}{\left|\text{leastK}^{(e)} \cup \text{leastK}^{(e^\star)}\right|}.
$$

We report $J_K(e, e+1)$ (adjacent checkpoints). Rising $J_K$ curves indicate that the identity–safe set becomes stable early.

**Rank stability of the full ordering.** Let $r^{(e)} \in \{1, \ldots, N\}^N$ be the rank permutation that sorts $\mathbf{d}^{(e)}$ ascending (1 = safest block to prune). We measure concordance with a reference ordering $r^{(e^\star)}$ via Kendall's $\tau$ and Spearman's $\rho$:

$$
\tau(e, e^\star) = \text{Kendall}\left(r^{(e)}, r^{(e^\star)}\right), \qquad \rho(e, e^\star) = \text{Spearman}\left(r^{(e)}, r^{(e^\star)}\right).
$$

Kendall's $\tau$ counts pairwise agreement; Spearman's $\rho$ is Pearson's correlation on ranks.

---

**Algorithm 1:** Post-Training Residual Block Replacement

---

**Input:** Pretrained ResNet model $M$, validation set $\mathcal{V}$, pruning threshold $\tau$
**Output:** Pruned model $M'$
Compute baseline accuracy: $\alpha_{\text{base}} \leftarrow \text{Accuracy}(M, \mathcal{V})$ ;
**for** *each residual block $B_i$ in $M$* **do**
    Temporarily ablate transformation: $F_i(x) \leftarrow 0$ ;
    Compute ablated accuracy: $\alpha_i \leftarrow \text{Accuracy}(M, \mathcal{V})$ ;
    Compute saliency: $\delta_i \leftarrow \alpha_{\text{base}} - \alpha_i$ ;
    Restore transformation: revert $F_i(x)$ ;

**for** *each residual block $B_i$* **do**
    **if** $\delta_i < \tau$ **then**
        Replace $F_i(x) \leftarrow 0$ (i.e., make block a skip) ;

---

### 3.5 ADVANTAGES AND DESIGN CONSIDERATIONS

The framework is post-training and architecture-preserving: it requires no sparsity regularizers and no channel surgery. Decisions are grounded in an interpretable, one-number diagnostic per block (Eq. 2). Finetuning after identity replacement is lightweight and experiment-dependent. Practically, identity replacements deliver hardware-visible latency/throughput gains while keeping the original topology, easing integration into existing inference pipelines.

## 4 EXPERIMENTAL SETUP

This section summarizes the choices that are shared across experiments. We describe the models and data we use, how we form fixed probe and eval splits, the checkpoints used to study training dynamics, and the computation of our diagnostic quantities ($\delta$ and magnitude proxies). We then detail how identity replacement and *finetuning* are applied, and conclude with the timing protocol for latency/throughput measurements.

**Datasets and splits.** We use ImageNet-1k and CIFAR-10. ImageNet evaluation uses the standard resize-shorter $= 256$ then $224{\times}224$ center crop; CIFAR-10 uses the conventional single-crop test with dataset mean/variance normalization.

**Models.** We report ImageNet results on ResNet-50 and training–dynamics on ResNet-101; CIFAR-10 uses ResNet-56. To test generality, we also evaluate ConvNeXt-Tiny and ViT-Tiny.

**Hardware.** All ImageNet experiments were run on 4×NVIDIA A100 (40 GB) GPUs with distributed data parallelism; CIFAR-10 runs used a single A100.

**Timing protocol.** We measure *GPU* forward latency/throughput with synthetic $(B, 3, 224, 224)$ inputs. Each run performs warmup steps. We report images/s and ms/batch as the median over $R$ repeats.

## 5 MAGNITUDE IS NOT NECESSITY

Magnitude-based heuristics are ubiquitous in pruning: large weights, BN scales, or activations are often treated as proxies for "importance" (Li et al., 2017; He et al., 2017; Liu et al., 2017; Lin et al., 2020; Molchanov et al., 2017; 2019). We test this claim in residual networks by correlating simple activation–magnitude proxies (Def. 3) with post-training block necessity (Eq. 2). All proxy scores are computed once on PROBE, a fixed, class-balanced 10% subsample of the training set, while necessity is always evaluated out-of-sample on the full validation set.

**Global aggregates can mislead.** Panel 2a shows an apparently strong positive association between output energy and when *all* blocks are pooled. Yet both axes shift systematically across stages

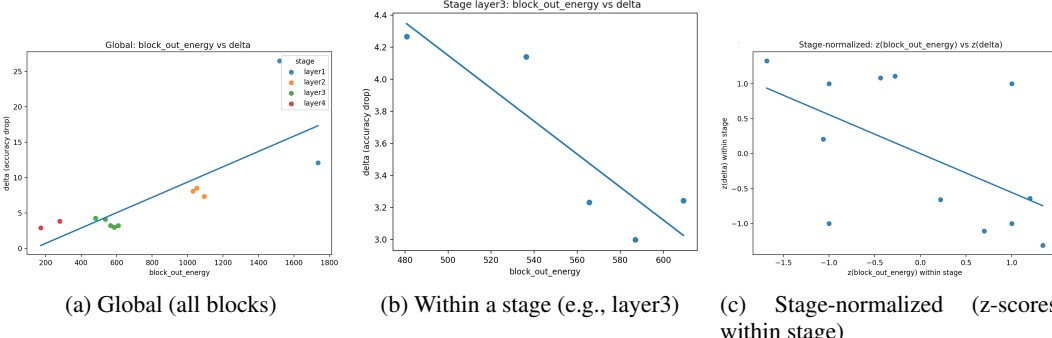

(a) Global (all blocks)      (b) Within a stage (e.g., layer3)      (c) Stage-normalized (z-scores within stage)

Figure 2: **Magnitude is not necessity.** (a) Aggregating over all blocks suggests a positive association between output magnitude and necessity ($\rho \approx 0.77$ in this run), but points cluster by stage. (b) Holding stage/depth fixed (e.g. layer3) the trend weakens or flips ($\rho \approx -0.70$ here). (c) After rank- or z-normalizing *within* stage and pooling, the association is consistently negative ($\rho \approx -0.56$): unusually large-magnitude blocks for their stage tend to be *less* necessary. Numbers vary slightly by seed; patterns are stable.

(earlier stages have larger spatial maps and channel widths), and the cloud decomposes into stage-specific clusters. The global statistic is therefore driven by *between-stage* scale differences rather than a consistent within-stage relationship.

**Within-stage, magnitude is a poor predictor.** Comparing like with like (Panel 2b) weakens or even reverses the apparent trend: within a fixed stage/depth, larger output magnitude does *not* reliably indicate greater necessity. To pool fairly across stages, Panel 2c z-scores each variable within its stage,

$$ z(v_i) = \frac{v_i - \mu_s}{\sigma_s} \quad \text{for block } i \in \mathcal{S}_s, $$

so points reflect how many stage-specific standard deviations a block lies above/below its peers. This removes coarse scale due to stage resolution/width. Under this normalization, the association remains weak and tends negative, reinforcing that raw magnitude is not a reliable indicator of necessity.

**Interpretation.** In skip-connected architectures, raw activation size can conflate *activity* with *novelty relative to the skip*. A block may emit large outputs that mostly mirror its input; such blocks appear "big" by magnitude yet contribute little by necessity. Hence, magnitude alone is an unreliable indicator for block selection under identity skips. It can be used as a complementary signal, but decisions should not rely on it without stage-/depth-aware controls and direct necessity ($\delta$) measurements.

## 6   REDUNDANCY EMERGENCE DURING TRAINING

We ask whether *which* residual blocks are safe to replace becomes *predictable* as optimization proceeds. At a set of saved checkpoints $\mathcal{E} = \{e_1, \ldots, e_T\}$ (ResNet-101: `e010`, `e030`, `e050`, `e070`, `e090`, `e100`), we compute per-block necessity on a fixed, class-balanced PROBE slice (10% of training): for block $i$ at epoch $e$, $\delta_i^{(e)}$ is the top-1 drop after identity-ablating that block (Eq. 2). Collecting $\boldsymbol{\delta}^{(e)} = \{\delta_i^{(e)}\}_{i=1}^{N}$ yields a time series of necessity maps (epoch × depth) and a full ranking of blocks at each checkpoint (ascending $\delta$).

**Stability metrics.** We quantify temporal stability in two complementary ways: (i) *full-order agreement* via Kendall's $\tau$ and Spearman's $\rho$ between the complete rankings induced by $\boldsymbol{\delta}^{(e)}$; (ii) *top-K identity* via the Jaccard index between the "least-$K$" sets leastK$^{(e)} = \arg \text{least}_K \boldsymbol{\delta}^{(e)}$ across adjacent epochs. The first asks whether the global ordering converges; the second asks whether the *identity* of the safest blocks locks in.

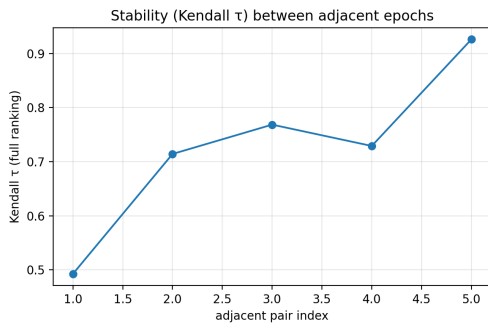
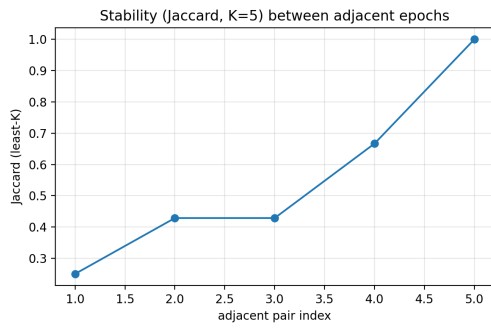

(a) Adjacent-pair Kendall's $\tau$ (full ranking).  (b) Adjacent-pair Jaccard (least-$K$, $K{=}5$).

Figure 3: **Ranking stability emerges over training.** (a) Full-order agreement rises through mid/late training. (b) Membership of the bottom-$K$ set stabilizes, indicating the identity of "safest-to-replace" blocks becomes predictable.

**Findings.** We observe clear emergence of structure. Full-order correlations (Fig. 3a; 4) increase from early to mid/late training, indicating convergence of the global necessity ordering. Concurrently, the Jaccard overlap of least-$K$ sets (Fig. 3b) rises and remains high by the last milestones, showing that the identity of the safest blocks stabilizes. A complementary $\delta$ heatmap (appendix) reveals spatial organization (e.g., low-$\delta$ concentration within mid/late `layer3`), while certain early/late blocks remain persistently high-$\delta$. Together, these results support the claim that training organizes residual capacity into a stable, stage-localized pattern, making one-shot post-training selection of least-$K$ blocks feasible and predictable.

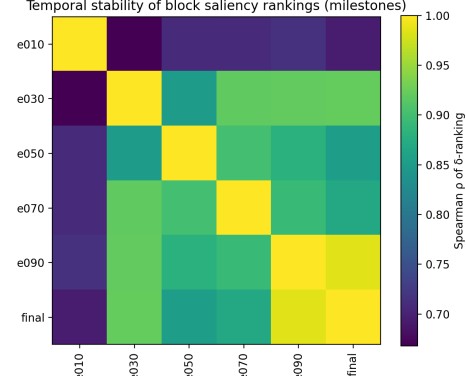

Figure 4: **Across-epoch agreement.** Spearman correlations between full $\delta$-based rankings at milestones show convergence by late epochs.

## 7 FROM ANALYSIS TO DEPLOYMENT

Stable saliency structure observed earlier makes the set of low-saliency residual blocks predictable by mid/late training. We turn this into a one-shot, post-training recipe: compute $\delta$ once, select the $K$ lowest-saliency residual blocks ("least-$K$"), replace their transformations by identity, and optionally run a short finetune—no channel surgery, no re-architecture. The emphasis here is on how the analysis *converts* into practical, deployment-oriented gains with minimal effort, rather than incremental leaderboard margins.

**Baseline sanity check (Table 2).** On ImageNet with ResNet-50, pruning a small number of residual blocks yields substantial, predictable efficiency gains for modest accuracy cost. For example, pruning $K{=}6$ cuts latency from $4.02\,\text{ms}$ to $2.68\,\text{ms}$ ($\approx 33\%$) and boosts throughput from 248 to 373 img/s ($\approx 50\%$) at only $0.5\,\text{pp}$ top-1 drop. Pushing to $K{=}7$ reaches $2.44\,\text{ms}$ and 410 img/s (about $39\%$ faster, $+65\%$ throughput) with a $1.2\,\text{pp}$ drop. These trends confirm that our analysis converts cleanly into real, controllable speedups.

Table 1: CIFAR-10 (ResNet-56). Accuracy change is relative to the unpruned baseline; **FLOPs reduction (%)**—*higher is better* ($\uparrow$).

| Method | Acc. $\Delta$ (%) | FLOPs $\uparrow$ (%) |
|---|---|---|
| HRank Lin et al. (2020) | $-2.5$ | 74.1 |
| ResRep Ding et al. (2021) | $-1.0$ | 77.8 |
| FPGM He et al. (2019) | $-0.3$ | 52.6 |
| DepGraph Fang et al. (2023) | $+0.1$ | 61.0 |
| Geometric He et al. (2020) | $-0.3$ | 41.0 |
| **Ours** | $-1.1$ | 73.8 |

**Positioning vs. structured pruning (Tables 1 and 4).** Relative to representative structured methods, our block-level least-$K$ offers competitive accuracy–efficiency with far less engineering. For instance, on CIFAR-10 (ResNet-56), our $73.8\%$ FLOPs cut at $-1.1\,\text{pp}$ sits in the same ballpark as

Table 2: Performance of our method on ResNet-50 (ImageNet) under varying pruning thresholds. Arrows indicate desirable direction of improvement.

| Pruned Blocks | Top-1 Accuracy (%) ↑ | Latency (ms) ↓ | Throughput (img/s) ↑ |
|---|---|---|---|
| 0 (Baseline) | 75.7 | 4.02 | 248 |
| 5 | 75.5 | 2.92 | 341 |
| 6 | 75.2 | 2.68 | 373 |
| 7 | 74.5 | 2.44 | 410 |

Table 3: ViT: Top-1 accuracy (%) with/without finetuning and parameter count (M) vs. pruned blocks $K$. Baseline shows the unpruned model. Finetuning uses **30 epochs**.

| K | Ours | | | LayerPrune | | |
|---|---|---|---|---|---|---|
| | No FT | FT | Params (M) | No FT | FT | Params (M) |
| 0 | 79.7 | N/A | 22.0 | 79.7 | N/A | 22.0 |
| 6 | 72.2 | 78.3 | 17.3 | 0.4 | 49.0 | 16.1 |
| 8 | 67.1 | 75.2 | 15.5 | 0.3 | 42.9 | 13.8 |
| 10 | 44.5 | 77.4 | 13.7 | 0.2 | 41.2 | 12.0 |

HRank (74.1%, $-2.5$ pp) and ResRep (77.8%, $-1.1$ pp) (Table 1). On ImageNet, $K=7$ achieves a 39.3% latency reduction with a 1.2 pp drop—on par with LayerPrune$_6$–Imprint's $\approx 40\%$ at 1.4 pp—while $K=6$ delivers 33.3% at just 0.5 pp (Table 4). The aim here is to highlight convertibility and deployment value, not incremental leaderboard margins.

**Beyond ResNets.** The same least-$K$ post-training recipe applies without modification to other residual-style architectures. On **ConvNeXt-Tiny** and **ViT-Small**, we compute $\delta$ once on a small PROBE slice, replace the $K$ lowest-saliency blocks with identity. Both models exhibit the same predictable accuracy–efficiency trade-offs under one-shot pruning and further gains with brief finetuning; see Table 5 and Table 6 (ConvNeXt), and Table 3 (ViT) for full results.

## 8  ROBUSTNESS AND SENSITIVITY

**How much probe data is needed?** A practical question for the diagnostic is how much probe data is required to recover a reliable block ordering. We recompute the per-block necessity ordering while shrinking the probe budget over a grid of ratios $\{1.0, 0.10, 0.05, 0.01\}$ and a class-balanced stratified split that uses *two samples per class*. The heatmap in Fig. 5 summarizes rank trajectories: rows are blocks (sorted by the full-budget reference), columns are budgets, and color encodes rank with stage boundaries overlaid. The picture is simple: the ordering is essentially unchanged across two orders of magnitude of ablation data; stage structure is preserved; visible differences are minor, within-stage swaps.

Table 4: Comparison of structured pruning methods on ResNet-50 with ImageNet. We report accuracy drop relative to each method's baseline and latency reduction with batch size 1. All latency values are from LayerPrune Elkerdawy et al. (2020).

| Method | Accuracy Drop (%) | Latency Reduction (%, bs=1)↓ |
|---|---|---|
| ThiNet Luo et al. (2017) | 4.1 | 10.8 |
| HRank Lin et al. (2020) | 4.2 | 11.9 |
| Channel pruning He et al. (2017) | 3.9 | 3.5 |
| LayerPrune$_6$–Imprint Elkerdawy et al. (2020) | 1.4 | 40.0 |
| LayerPrune$_2$ Elkerdawy et al. (2020) | 0.3 | 30.8 |
| **Ours (5 blocks)** | **0.2** | **27.4** |
| **Ours (6 blocks)** | **0.5** | **33.3** |
| **Ours (7 blocks)** | **1.2** | **39.3** |

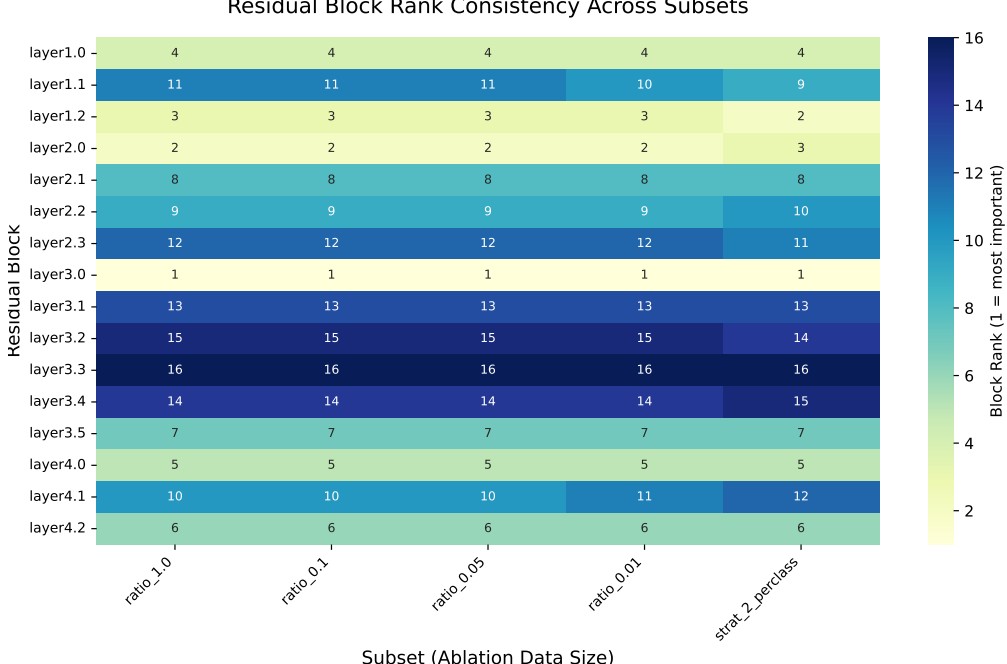

Figure 5: **Ranking stability emerges over training.** Spearman correlations between $\delta$-based block rankings at milestones.

**Quantifying stability.** We compare the *rank vectors* at each reduced budget to the full-budget reference using Pearson correlation. Agreement is effectively perfect at $10\%$, $5\%$, and $1\%$ ($r = 1.00$ in all three) and remains very high for the stratified split ($r = 0.97$). Together with the visual evidence, these results support the lightweight-probe design: a small, fixed probe slice suffices to recover the salient $\widehat{\delta}$ structure, while keeping compute modest when sweeping checkpoints or architectures.

## 9 CONCLUSION

We examined residual networks through three lenses and turned the insights into a simple recipe. (i) *Magnitude vs. necessity:* large norms are unreliable; $\delta$-based ablations better capture block necessity. (ii) *Emergence of redundancy:* saliency rankings stabilize by mid/late training, preserve stage structure, and remain robust even with $1\%$ probe data. (iii) *Pruning structure:* replacing the $K$ least-salient blocks with identity (least-$K$) is a post-training, architecture-preserving step requiring only a single saliency pass and an optional short finetune.

Across ImageNet (ResNet-50, ConvNeXt-Tiny and ViT-Small) and CIFAR-10 (ResNet-56), this converts analysis into predictable efficiency gains (lower latency / FLOPs) at small, controllable accuracy costs, while staying compatible with standard inference engines and avoiding channel surgery.

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

# APPENDIX

## A CONVNEXT-TINY: PREDICTABLE POST-TRAINING PRUNING

We apply the same least-$K$ recipe to **ConvNeXt-Tiny**: compute $\delta$ once on a 10% PROBE slice, rank blocks, and replace the $K$ lowest-saliency blocks with identity. Evaluation uses the full ImageNet-1k validation set.

Table 5: Accuracy and parameter count vs. number of pruned blocks $K$ under **one-shot pruning (no finetuning)**. Top-1 in %, Params in millions (M).

| Method | Metric | K=3 | K=5 | K=7 |
|---|---|---|---|---|
| Ours | Top-1 (%) | 79.2 | 73.0 | 54.0 |
| | Params (M) | 26.7 | 24.3 | 23.0 |
| LayerPrune | Top-1 (%) | 78.3 | 67.3 | 53.2 |
| | Params (M) | 24.9 | 23.5 | 23.3 |

Table 6: ConvNeXt: finetuned accuracy (%). Only 30 epochs of finetuning were used for all runs.

| K | Acc. (LayerPrune) | Acc. (Ours) |
|---|---|---|
| 3 | 78.9 | 80.7 |
| 5 | 78.5 | 78.4 |
| 7 | 77.2 | 77.0 |

## B    HARDWARE AND EVALUATION SETUP

**Inference Benchmarking Protocol.**    To evaluate latency and throughput, we measured single-image inference performance using synthetic inputs on both CIFAR-10 and ImageNet. All benchmarks were conducted using a batch size of 1 to simulate real-time, low-latency inference scenarios. Latency was measured using Python wall-clock timing with `torch.cuda.synchronize()` to ensure accurate GPU timing.

Each measurement included:

- 30 warm-up iterations (not timed) to stabilize kernel selection via cuDNN autotuning.
- 100 timed forward passes for performance averaging.
- 100 independent benchmarking repeats to compute min, max, mean, and median statistics.

**Throughput and Latency Reporting.**    All throughput values are reported in images per second (img/s), while latency is reported in milliseconds per image (ms/img). The final reported numbers in the main paper correspond to the *median* values across the 100 benchmarking repeats for each model configuration.

## C    QUALITATIVE COMPARISON OF PRUNING METHODS

We provide a qualitative comparison of structured pruning methods evaluated on ResNet-50 with ImageNet. The **Type** column indicates whether pruning is applied during training or as a post-training step, while the **Iterative** column reflects whether the method relies on progressive, multi-step pruning. Our method stands out for its simplicity: a non-iterative, post-training approach that requires no retraining schedule or specialized optimization objectives.

Table 7: Overview of pruning methods evaluated on ResNet-50 with ImageNet. We report pruning granularity, application stage, and whether the method is applied iteratively.

| Method | Granularity | Type | Iterative |
|---|---|---|---|
| HRank Lin et al. (2020) | Channel | During-Train | ✓ |
| ResRep Ding et al. (2021) | Channel | During-Train | ✓ |
| FPGM He et al. (2019) | Channel | Post-Train | − |
| DepGraph Fang et al. (2023) | Channel+Group | During-Train | ✓ |
| Geometric He et al. (2020) | Channel | During-Train | − |
| Channel pruning He et al. (2017) | Channel | During-Train | ✓ |
| DepthShrinker Fu et al. (2022) | Block | During-Train | ✓ |
| LayerPrune (Imprint) Elkerdawy et al. (2020) | Block | Post-Train | − |
| ThiNet Luo et al. (2017) | Channel | Post-Train | ✓ |
| **Ours** | Block | Post-Train | − |

## D    TRAINING AND FINETUNING DETAILS

**Optimizer and Loss.**    For both CIFAR-10 and ImageNet experiments, finetuning was performed using the AdamW optimizer with standard cross-entropy loss. Weight decay regularization was applied, using values tuned for each dataset.

**CIFAR-10 Setup.**    We finetuned the pruned ResNet-56 models on CIFAR-10 for 30 epochs with a batch size of 256. The learning rate was set to 0.001 and scheduled using cosine decay with linear warm-up over the first 10% of epochs. A weight decay of 0.05 was used. All model parameters were unfrozen during finetuning, and training was conducted in full-precision (FP32) without mixed-precision acceleration. Data augmentation included random cropping with 4-pixel padding and horizontal flipping, followed by normalization using the standard CIFAR-10 statistics.

**ImageNet Setup (ResNet-50).**   For ImageNet, pruned ResNet-50 models were finetuned for 100 epochs using 4 NVIDIA A100 GPUs in distributed data parallel mode. Each GPU processed a batch size of 256, resulting in an effective batch size of 1024. The learning rate was set to 0.0005 with cosine decay and a 10% linear warm-up phase. A weight decay of 0.1 was used. As with CIFAR-10, all layers were trainable and no early stopping or learning rate restarts were applied. Training was done using full-precision computation.

**ImageNet Setup (ViT).**   For ViT we finetune pruned DeiT-S/16 for 30 epochs in multi-GPU DDP (per-GPU batch size 256; learning rate scales with global batch as $\text{lr}_{\text{eff}} = \text{lr} \times \frac{\text{global\_batch}}{256}$; e.g., $1.25 \times 10^{-4} \rightarrow 5 \times 10^{-4}$ for 4×256). We use AdamW (weight decay 0.05), cosine decay with a learning-rate floor ($\text{lr}_{\min} = 1 \times 10^{-5}$, scaled with global batch), and *no warmup*. Label smoothing is 0.1. Layer-wise learning-rate decay is enabled (decay 0.75) with a ×10 head LR multiplier; no weight decay on norms/bias/pos_embed. Stochastic depth is disabled; models run in channels-last format. Mixup/CutMix are enabled via `timm` and ramped down over the final 20% of epochs. An exponential moving average (EMA, decay 0.9999) is maintained and applied for validation.

**ImageNet Setup (ConvNeXt-Tiny).**   We finetune pruned ConvNeXt-Tiny for 30 epochs in multi-GPU DDP with per-GPU batch size 256 (global batch scales the learning rate as $\text{lr}_{\text{eff}} = \text{lr} \times \frac{\text{global\_batch}}{256}$; in our runs $\text{lr} = 1.0 \times 10^{-4}$ at global batch 256). We use AdamW (weight decay 0.05), a cosine schedule (no warmup; warmup ratio 0.0), mixed-precision (AMP), and standard ImageNet preprocessing (224×224, bicubic resize to 256, center crop, mean/std normalization). Models run in channels-last format; label smoothing is 0.0. An EMA of weights (decay 0.999) is maintained and applied for validation.

**Evaluation.**   Model selection and reporting were based solely on top-1 accuracy. Evaluation was performed on the full validation set using `torch.no_grad()` mode, with no test-time augmentation or ensembling.

# E   SALIENCY ANALYSIS ACROSS DATASETS

Figure 6 shows block-wise saliency scores for ResNet-101 on CIFAR-10 and ImageNet. We observe a consistent trend across both datasets: the first block of each stage (e.g., `layer2.0`, `layer3.0`, `layer4.0`) tends to be highly salient, highlighting its structural importance. At the same time, differences in dataset complexity reveal how overparameterization affects redundancy. CIFAR-10, being a simpler task than ImageNet, allows clearer identification of blocks that contribute minimally—such as `layer3.7` through `layer3.9`—where disabling the main path has almost no impact on accuracy. These observations reinforce the utility of our method in revealing interpretable patterns of redundancy, particularly when scaling large models to less demanding tasks.

## ETHICS STATEMENT

We adhere to the ICLR Code of Ethics. This work studies post-training simplification of standard vision backbones using publicly available datasets (ImageNet-1k, CIFAR-10) under their respective licenses. It does not involve human subjects, personal or sensitive data, or demographic attributes, and we are not aware of ethical, legal, or privacy concerns arising from the methodology. As with any pruning technique, downstream deployments should re-evaluate accuracy (and potential shifts across subpopulations) on the target distribution and comply with local data-use policies. We disclose no conflicts of interest or external sponsorship influencing the results.

## REPRODUCIBILITY STATEMENT

The experimental setup and assumptions (datasets, model variants, evaluation protocol) are described in §4; robustness and probe construction are detailed in §8; and full experiment settings (seeds, split definitions, and pruning/finetuning schedules) are collected in App. D. At submission time we are not releasing code; at camera-ready we will provide an evaluation package (environ-

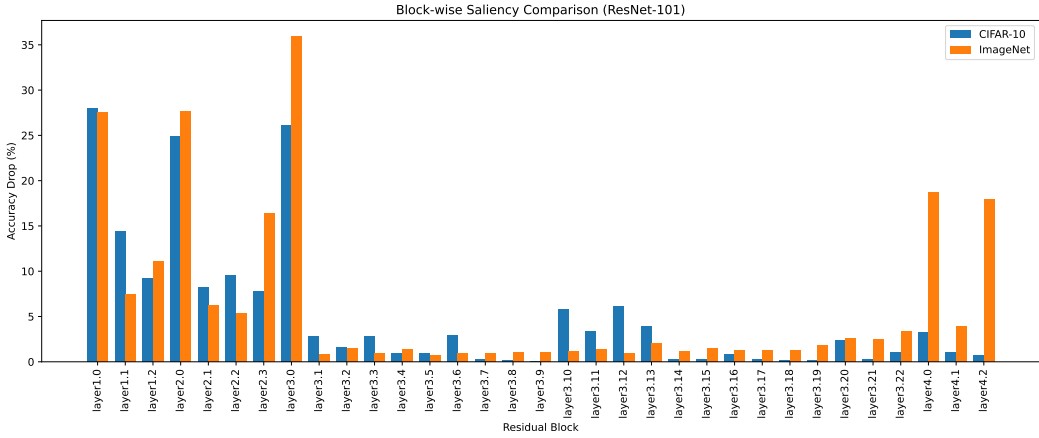

Figure 6: Block-wise saliency scores for ResNet-101 on CIFAR-10 and ImageNet. Each bar represents the drop in top-1 accuracy when the main path of the corresponding residual block is disabled. Our method reveals consistent structural redundancy in later blocks across datasets.

ment specification, scripts, and precomputed saliency scores / least-$K$ indices) referenced from the supplement, along with a README that maps each command to the tables and figures in the paper.

