# OpenReview forum: "Where Redundancy Lives: Stage-Aware Block Saliency in Skip-Connected Models"
_ICLR.cc/2026/Conference — ICLR 2026 Conference Withdrawn Submission_

### Official Review · Reviewer_oZ4i · 2025-10-26

**Soundness:** 3
**Presentation:** 3
**Contribution:** 2
**Rating:** 2
**Confidence:** 5

**Summary:**

The paper probes redundancy in residual networks by identity-ablating blocks, ranking them via accuracy drop, and pruning the least “necessary.” Experiments cover ResNet-50/101 on ImageNet, ResNet-56 on CIFAR-10, and brief results for ConvNeXt-Tiny and ViT-Tiny; the paper also reports latency/throughput trade-offs after identity replacement.

**Strengths:**

- The proposed diagnostic approach is simple yet effective, preserving the original network architecture without requiring any structural modifications such as channel pruning or layer surgery. Its straightforward implementation makes it easily applicable across various model families.
- The stage-aware analysis provides a principled framework for understanding the relative importance of different blocks within the network. Accounting for hierarchical dependencies across stages, it offers clearer interpretability into how architectural components contribute to overall performance.
- Empirical results demonstrate consistent and predictable trade-offs between accuracy and latency.

**Weaknesses:**

- The proposed probe–score–prune pipeline bears a strong resemblance to existing block pruning and dynamic skipping frameworks such as LayerPrune [1], BlockDrop [2], and SkipNet [3]. While the implementation is clean and the framing is coherent, the conceptual advance over these prior works appears incremental.
- The experimental comparisons rely heavily on older methods and report latency numbers drawn from heterogeneous hardware setups. A consistent, same-device evaluation against more recent approaches from 2022–2025 would substantially strengthen the empirical credibility.
- The analysis primarily employs simple magnitude-based importance proxies, without testing stronger alternatives such as curvature- or gradient-based criteria. Furthermore, results are mostly presented as single-seed point estimates, leaving questions about robustness and statistical significance.
- The experiments focus predominantly on ResNet-style architectures, with only brief coverage of ConvNeXt and ViT models. The absence of modern training protocols and larger-scale backbones limits the generality and practical relevance of the findings.

## Reference:
- [1] To Filter Prune, or to Layer Prune, That Is The Question. ACCV2020
- [2] BlockDrop: Dynamic Inference Paths in Residual Networks. CVPR2018
- [3] SkipNet: Learning Dynamic Routing in Convolutional Networks. CVPR2018

While the paper is clearly written and technically sound, its motivation and contribution are not sufficiently compelling given the current state of the literature. The topic of block importance and pruning diagnostics has been extensively explored in prior work, and the empirical results presented here do not demonstrate substantial improvements or new insights. Consequently, it is unclear why this line of investigation warrants further attention at this stage. In its current form, the paper falls short of the novelty and impact expected for acceptance at a top-tier venue such as ICLR.

**Questions:**

- How does the proposed ranking method perform relative to established importance estimation techniques such as Fisher information, Taylor expansion, Hessian-based sensitivity, or influence-function–derived scores? A direct quantitative comparison would help clarify whether the proposed diagnostic offers distinct advantages beyond simplicity.
- Could the authors provide uniform, same-hardware measurements of latency and throughput when comparing against recent block-pruning and dynamic routing baselines? Such controlled evaluations would eliminate confounding factors and offer a clearer view of the true efficiency–accuracy trade-offs.

---

### Official Review · Reviewer_HeYp · 2025-10-28

**Soundness:** 2
**Presentation:** 1
**Contribution:** 1
**Rating:** 2
**Confidence:** 5

**Summary:**

This paper proposes to use the saliency score to identify redundant blocks in skip-connected models. The saliency score is defined as the
accuracy drop (on a small training set) if a residual block is dropped. The authors find that magnitude based block/layer pruning is an inconsistent predictor of saliency score. And the saliency score stabilizes at the later stages of the model training. Overall, the proposed saliency score based block pruning method is one-shot, and achieves good performance after finetuning.

**Strengths:**

The saliency score based block pruning method is simple and easy to use.

**Weaknesses:**

1. This paper is hard to follow. The presentation should be improved.
2. The core contributions are limited. The conclusions summarized in this paper are well known or trivial‌. The $\delta$ importance score is simple and previously studied, eg. [1]. Magnitude based metric is not consistent with the final performance, especially after finetuning. These are also well known. The proposed $\delta$ importance score is also not optimal when multiple layers are considered, or finetuning is applied.
3. Comparison with SOTA methods are missing, for example [2]. The results didn't show significant improvement of the proposed method, even compared with LayerPrune in table 4.

[1] Pruning Filters for Efficient ConvNets. ICLR 2017.
[2] Effective Layer Pruning Through Similarity Metric Perspective. ICML 2024.

**Questions:**

What is the main contribution of this method? Is it the overall pruning pipeline, or the importance score metric?

How is the results compared with more recent sota method? How the proposed method works on larger models like LLMs?

---

### Official Review · Reviewer_cbXW · 2025-11-01

**Soundness:** 3
**Presentation:** 3
**Contribution:** 1
**Rating:** 2
**Confidence:** 4

**Summary:**

The paper studied how to replace redundant blocks with skip connections. The paper replaced a skip connection block y = f(x) +x with just y= x, to measure change in accuracy and then remove unimportant blocks. The paper follows a typical prune-finetune strategy. The paper performed an experiment to show that high magnitude is not necessary for pruning, (as skip connections carry forward magnitude from previous layers, its unreliable). The paper performs experiments on Imagenet on ResNet50, VIT-tiny and Convnext to assert the claims.

**Strengths:**

1. The paper is well structured and easy to follow.
2. The claims that a). Magnitude is unreliable for pruning skip connected blocks b) Finding the actual contribution of the main block is better metric of importance, are valid and the experiments that show this are properly formulated and presented.
3. The paper proposes that a better metric to assess importance of blocks is to assess the novel contribution of each block, instead of considering the collective magnitude, which is true when we consider the block always has a skip connection, which is justified.

**Weaknesses:**

1. The main issue is novelty, There are multiple prior works that formulate replacing skip connections with identity blocks for pruning[1][2], There are also papers which discuss subnetwork stability in neural networks as training progresses [3] which also applies to importance of residual blocks. There are also works that discuss that magnitude != importance[4]. Therefore, though I find the experiments interesting, its not novel enough for publication.
2. The experimentation is performed on small models, trying on larger models like the DeiT Base can give better analysis and proof of phenomenon discussed.
3. Skip Connections are widely used for Language Tasks, some experiments applying the analysis to text data would be  insightful.




References:
[1] Zuxuan Wu, Tushar Nagarajan, Abhishek Kumar, Steven Rennie, Larry S. Davis, Kristen Grauman, Rogerio Feris; Proceedings of the IEEE Conference on Computer Vision and Pattern Recognition (CVPR), 2018, pp. 8817-8826
[2] Wang, Xin, et al. "Skipnet: Learning dynamic routing in convolutional networks." Proceedings of the European conference on computer vision (ECCV). 2018.
[3] Frankle, Jonathan, and Michael Carbin. "The lottery ticket hypothesis: Finding sparse, trainable neural networks." arXiv preprint arXiv:1803.03635 (2018).
[4] Molchanov, P., Tyree, S., Karras, T., Aila, T., & Kautz, J. (2017). Pruning Convolutional Neural Networks for Resource Efficient Inference. International Conference on Learning Representations. Retrieved from https://openreview.net/forum?id=SJGCiw5gl

**Questions:**

refer weakness, no explicit questions.

---

### Official Review · Reviewer_CYi4 · 2025-11-01

**Soundness:** 3
**Presentation:** 3
**Contribution:** 2
**Rating:** 4
**Confidence:** 3

**Summary:**

This paper provides an interesting, empirical approach toward identifying which residual blocks can be pruned from models with skip connections. Unlike structured pruning techniques, which remove individual filters, this work focuses on entire blocks of residual networks. The proposed method involves replacing residual blocks solely with identity maps, and measuring the resulting change in accuracy. The authors also show that standard measures of block importance are inconsistent. They show promising results on compressing a variety of models, including ResNets, ConvNexts, and ViTs.

**Strengths:**

S1. The notion of residual blocks with identity maps reduces model size and improves inference time. The authors proposed method is a clear and interpretable way to identify which blocks to remove or keep.

S2. The proposed method essentially retains the key features of the model's architecture, without requiring extensive modifications that arise when more fine-grained pruning (such as structured pruning) are used.

S3. The empirical study is quite thorough, and shows uniform improvement over near baselines (though the improvements are typically incremental).

S4. The analyses showcasing relative importance of blocks within stages, and the stages themselves, is of independent interest and could be useful for other compression techniques as well.

**Weaknesses:**

W1. While the paper is generally well written, there are a few writing issues that could be cleared up
* In the introduction $\alpha$ values are not defined
* When the probe and eval datasets are defined, they are written differently (Capitalized when defined, in normal text in the paragraph, and in small caps in the equations). This should be constant across measures.

W2. A few minor issues with the related work - for structured pruning, other methods such as DFPC (Narshana et al, 2023), and GroupFisher (Liu et al, 2021) should be mentioned and if possible, baselined against. These are papers that explicitly address the problem of structurally pruning models possessing skip connections (in the parlance of Table 6, they would fall under "Channel+Group"), and it would be interesting to see how block pruning compares with them (beyond the works already cited).

W3. The saliency methods used for comparision are generally quite narrow in scope. It would be interesting to see how the analyses provided in sections 3 and 5 would play out with, say, Gradient/Taylor series based scores or discrimination-based scores.

**Questions:**

Please refer to Weaknesses section.

---

### Note · Authors · 2025-11-13

I have read and agree with the venue's withdrawal policy on behalf of myself and my co-authors.